# Evaluation of Microalgae Antiviral Activity and Their Bioactive Compounds

**DOI:** 10.3390/antibiotics10060746

**Published:** 2021-06-20

**Authors:** Dora Allegra Carbone, Paola Pellone, Carmine Lubritto, Claudia Ciniglia

**Affiliations:** 1Department of Environmental, Biological and Pharmaceutical Sciences and Technologies, University of Campania “Luigi Vanvitelli”, Via Vivaldi 43, 81100 Caserta, Italy; carmine.lubritto@unicampania.it (C.L.); claudia.ciniglia@unicampania.it (C.C.); 2Department of Marine Biotechnology, Stazione Zoologica Anton Dohrn, Villa Comunale, 80121 Naples, Italy; paola.pellone@gmail.com; 3National Institute of Nuclear Physics, Complesso Universitario di Monte S, 80126 Naples, Italy

**Keywords:** microalgae, virus, antiviral compounds, SARS-CoV-2, microalgae antiviral activity, coronavirus disease

## Abstract

During the last year, science has been focusing on the research of antivirally active compounds overall after the SARS-CoV-2 pandemic, which caused a great amount of deaths and the downfall of the economy in 2020. Photosynthetic organisms such as microalgae are known to be a reservoir of bioactive secondary metabolites; this feature, coupled with the possibility of achieving very high biomass levels without excessive energetic expenses, make microalgae worthy of attention in the search for new molecules with antiviral effects. In this work, the antiviral effects of microalgae against some common human or animal viruses were considered, focusing our attention on some possible effects against SARS-CoV-2. We summed up the data from the literature on microalgae antiviral compounds, from the most common ones, such as lectins, polysaccharides and photosynthetic pigments, to the less known ones, such as unidentified proteins. We have discussed the effects of a microalgae-based genetic engineering approach against some viral diseases. We have illustrated the potential antiviral benefits of a diet enriched in microalgae.

## 1. Introduction

Viruses are the smallest and the most common entities on Earth. Their uniqueness lies in the fact that they are able to multiply only inside the cells of other living things. Viruses are generally considered non-living, since they are made up of a core of genetic material, either DNA or RNA, surrounded by a protective protein coat called a capsid [1].

Viruses have had evolutionary success for four main reasons: a massive genetic variation, a variety in means of transmission, a good mechanism to replicate in the host and the ability to survive in the host [2] A viral infection is generally divided into three steps [3]: (*i*) the virus binds to cell receptors on the cell membrane, transferring its genome into the cytoplasm—this process allows the virus to integrate into the host’s genome; (*ii*) the cells are forced to reproduce the virus genome; (*iii*) the protein and virion progeny are produced.

Viruses cause several human infections [4]; the so-called severe acute respiratory syndrome is caused by a coronavirus (SARS-CoV). One of these infections has recently dramatically risen to prominence because of its level of lethality along with the high degree of contagiousness of its etiological agent: severe acute respiratory syndrome corona virus 2 (SARS-CoV-2) [5,6].

SARS-CoV-2 is a spherical, enveloped single-stranded RNA virus of the *Coronaviridae* family. The coronavirus spike (S) protein binds angiotensin-converting enzyme 2 (ACE2) receptors, located on the surfaces of many human cells, allowing virus entry and replication, and consequently the development of the disease [6]. This virus appeared at the end of 2019 in Wuhan (China), spread worldwide, and is now considered a pandemic by the World Health Organization, causing many deaths and indirectly also the downfall of the world economy.

The main weapons to prevent viral diseases are vaccines; for some viruses, such as human immunodeficiency virus (HIV), Zaire ebolavirus (ZEBOV), and Dengue virus (DENV), vaccines are still not available [7,8]. Moreover, some steps of virus replication take place inside the cellular metabolic pathway; therefore, it is not easy to treat the virion. However, viruses have particular targets present only in their structure, on which it is possible to act [2].

Some antiviral compounds have natural origins, and microalgae are among the most interesting candidates for their production. Among marine biomedical compounds, 9% are isolated from algae [9,10], and some of these products are composed of molecules that are currently impossible to reproduce by chemical synthesis [11,12,13]. Moreover, microalgae have the ability to achieve very high biomass levels, and consequently high compound products, without excessive energy costs [14,15,16]. 

One of the first studies on the antiviral activity of microalgae was carried out by Umezawa et al. [17]: an extract containing acid polysaccharides from *Chlorella pyrenoidosa* was shown to have an inhibitory effect in mice against vesicular stomatitis virus (VSV). However, the greatest interest in the antiviral compounds within microalgae spread only in the mid-1990s, when some cyanobacteria, such as Nostoc and Spirulina, showed interesting results against HIV, which was considered a plague in the 1980s [17,18,19,20,21].

Recently, the interest in microalgae with antiviral effect has also developed in one of the biggest economic sectors, that of aquaculture and its derivatives [22]. Indeed, water is an excellent vector for the transmission of viruses that often kill or seriously damage some animals [23].

In this review, we analyzed the antiviral effects of microalgae against some common human and animal viruses. Moreover, we examined antiviral compounds produced by microalgae, the effect of genetic engineering in this field, and perspectives on a diet enriched in microalgae. We focused our attention on some possible effects against SARS-CoV-2.

## 2. Microalgae Antiviral Compounds

### 2.1. Lectins

Among the different strategies to fight viruses, one involves the prevention of the entry of the microbes into the cells by blocking their interaction with the cell membrane. The most famous preventive agents are lectins, proteins of a non-immunoglobulin nature capable of binding carbohydrates without altering the structures of specific glycosyl ligands; for this reason, they are called carbohydrate-binding agents (CBAs) [24]. Lectins have internal repeats containing a carbohydrate recognition domain (CRDs) and they are classified via this domain.

The orientation of CRDs determines the affinity of the carbohydrate structure. Therefore, some lectins only interact with high-mannose polysaccharides, some with glycan branches, and others with the galactose core [24]. Lectins interact directly with the high-glycan structure of virally enveloped glycoproteins that are added after post-translational modification [25].

Microalgae lectins with antiviral activity all originate from cyanobacteria, and all of them have anti-HIV activity (Table 1; Figure 1).

HIV is a positive single-stranded enveloped RNA virus of the genus *Lentivirus* of the *Retroviridae* family that infects CD4^+^ cells. This virus induces the lowering of CD4^+^ cell levels, making the body more vulnerable to all types of infections and causing acquired immunodeficiency syndrome (AIDS). There are two main types of this virus: HIV1, diffused around the world, and HIV2, diffused only in West Africa because of its low transmissibility [26,27].

In more detail, HIV enters the cells thanks to the envelope protein gp120 that engages the CD4^+^-like receptor and the CCR5- or CXCR4-like coreceptor. After this interaction, the gp120 and gp41 peptides (glycosylated trimeric assemblies interacting with lectins) fuse inside the membrane [28,29] (Figure 1).

There are five lectins from cyanobacteria that are the most studied: Agglutinin OAA, Cyanovirin-N (CV-N), Microcystis Viridis Lectin (MVL), Microvirin, and Scytovirin (Table 1). 

*Oscillatoria agardhii* strain NIES-204 produces a lot of lectins to support blooms on the surfaces of lakes and ponds; among these, there is Agglutinin OAA (molecular weight 13.9 kDa). This lectin shows anti-HIV1 and anti-HIV 2 activity in vitro, and high mitogenic activity [30,31,32,33].

CV-N was extracted from *Nostoc ellipsosporum* [34,35] and clinical trials around this substance are very advanced: more than 200 papers have been published in the last twenty years [20,36,37,38,39,40]. In the USA, it has become a commercial drug, produced by Cellegy Pharmaceuticals Inc., and in experiments and different clinical trials, it has been used as a rectal and vaginal gel [40].

Different studies have shown that CV-N inhibits the entry into ocular human cells of three viruses: hepatitis C virus (HCV), ZEBOV [41] and herpes virus simplex (HSV) [42,43]. 

HCV is an enveloped positive RNA virus from the Flaviviridae family that replicates on intracellular lipid membranes in hepatocytes, but also in peripheral blood mononuclear cells [44]. 

ZEBOV is a negative-stranded virus from the Filoviridae family causing a hemorrhagic fever with a high mortality rate [45]. 

HSV is a double-stranded DNA enveloped virus of the *Herpesviridae* family causing lithic infections in human mucoepithelial cells, such as fibroblasts and epithelial cells. After the infection, the virus moves to sensory nerve axons and causes latent infection in trigeminal ganglia, thanks to some genes that control the latency-associated transcript. Therefore, it can show up again periodically. The two main forms, HSV1 and HSV2, respectively cause oral and genital herpes [46]. 

CV-N also contains anti-influenza A-B viruses (IAV, IBV). They are negative stranded RNA viruses belonging to *the Orthomyxo-viridae* family. IAV is the only species of the genus *Alphainfluenzavirus*, and is categorized into subtypes based on the type of each of the two proteins present on the viral envelope, hemagglutinin (H) and neuramidase (N) (HxNy). IBV is the only species of the genus *Betainfluenzavirus* [47]. Viruses belonging to this family have the ability to adsorb the glycoprotein receptors of the red blood cells, generally causing acute respiratory infections, which are highly contagious. This is called “influenza” in humans and animals [48].

CV-N binds neuraminidase, opening a gap in the cell membrane [49]. Generally, CV-N is expressed in different organisms and carried out different actions, e.g., *Escherichia coli*, *Streptococcus gordonii*, *Lactobacillus Jensenii* [50,51,52,53,54]. 

Microvirin is an alpha-(1,2)-mannose-specific lectin; it is a protein of 14.7 kDa and shares 33% amino acid identity with CV-N [33]. It is produced by *Microcystis aeruginosa* PCC7806 in higher quantities in the presence of iron and light stress conditions [33,55]. Generally, it is expressed thanks to *Escherichia coli*, and has a good anti-HIV activity, helping to avoid syncytium formation. It has a higher safety profile, but it is not possible to achieve a high productivity level of CV-N [56,57].

MVL is a protein of 13 kDa containing two carbohydrate-binding sites per monomer. It is produced by the PCC7806 strain of *Microcystis viridis* [58]. As for Agglutinin OAA, data suggest that MVL can be toxic due to its interactions with cellular proteins [56,59,60]. Kachko et al. [43] showed, thanks to biolayer interferometry, that this lectin also has anti-HCV activity, having high affinity with the HCV E1E2 glycoprotein. Scytovirin is produced by *Scytonema varium*, and it has a weight of 9.71 kDA and a high affinity with tetrammonose. In vitro, its N-terminal residue of the SD1 domain links to HIV1 [61,62]. Western blot analysis confirms that this lectin has some effect against ZEBOV and HCV, too [63]. In more detail, Garrison et al. [45] showed that in some 9,BALB/c mice infected with ZEBOV, the presence of this lectin reduced mortality, and its action had some effects when the lectin was added from the day before the infection until the day after the infection.
antibiotics-10-00746-t001_Table 1Table 1Microalgae lectins with antiviral effects.Lectins Interact Directly with the High-Glycan Structure of Viral Envelope Glycoproteins**Name****Organism****Virus****References**Agglutinin OAA*Oscillatoria agardhii*HIV1, HIV2[31,32,33]CV-N*Nostoc ellipsosporum*HIV1, HSV1, HCV, IAV, IBV[35,36,39,40,41,42,50,52]Microvirin*Microcystis aeruginosa*HIV1, HIV2[54,55]MVL*Microcystin viridis*HIV1, HIV2, HCV[57,58,59,60]Scytovirin*Scytonema varium*HIV1, ZEBOV[61,62,63]

### 2.2. Polysaccharides

Polysaccharides originate in a great variety of organisms. Among these substances, sulphate polysaccharides and acidic polysaccharides showed the highest antiviral activities.

Sulphate polysaccharides have a high antiviral activity against enveloped viruses. They interact with the positively charged domains of the virus glycoprotein envelope and create a non-reversible complex; as such, they occupy the sites for virus attachment (Figure 2).

The activity of sulphate polysaccharides is linked to different factors: the degree of sulphation, molecular weight, the distribution of sulphate in the structure and stereochemistry, the effect of counter cations, and hydrophobic and hydrogen bonding interaction [64,65,66,67,68].

Among the different microalgae producing sulphate polysaccharides, the best candidates are definitely *Spirulina* and *Porphyridium* (Table 2).

The most famous sulphate polysaccharide from *Spirulina* is Calcium-spirulan (Ca-SP), which was isolated by Hayashi and Hayashi in 1996 from a hot water extract via cellulose chromatography. Ca-SP showed its activity against HIV1 by interacting with the V3 loop region of the viral envelope, containing gp120, and stopping the syncytium formation between infected cells and non-infected cells [4,69,70]. This polysaccharide also prevents syncytium formation in the presence of HSV1 and HSV2 in vitro and in vivo [69].

Lastly, this compound also acts against other enveloped viruses, such as human cytomegalovirus (HCMV), and Mumps virus (MuV), IAV [69].

HCMV is a double-stranded DNA virus of the *Herpesviridae* family, showing its action against people with compromised immune systems and causing retinitis, pneumonitis, and possible liver failure [71]. MuV is a single-stranded RNA virus of the *Paramyxoviridae* family causing pain in the salivary glands [71,72].

Mass spectrometry showed that Ca-SP is composed of two types of repeating units of disaccharides (O-rhamnosyl-acofriose and *O*-hexuronosyl-rhamnose). It has a high antiviral activity against all enveloped viruses thanks to the chelation of calcium ions with a sulphate group [19,66].

Radonic et al. [72] showed that *Spirulina platensis* also produces TK-V3, another sulphate polysaccharide. In this study, TK-V3 in vitro reduced the replication of two animal viruses, vaccinia virus (VV) and Ectromelia virus (ECTV), but it also has some effect against HSV1. VV is a double-stranded DNA-enveloped virus of the Poxviridae family, and is a constituent of the vaccine that eradicated smallpox and caused smallpox in cattle [71]. ECTV is an enveloped double-stranded DNA virus of the Poxviridae family that causes mousepox in mice [72].

*Arthsospira fusiforme* produces a sulphate polysaccharide with interesting anti-HSV1 and HSV2 activity, and is thought to be linked to the presence of phycobiliproteins [73].

*Porphyridium* is a red microalga of the Porphyridiophyceae family that, in recent years, has received great attention for its possible biotechnological uses in the biomedical field. This microalga is encapsulated by an envelope of sulphate polysaccharides with antitumor, antibacterial and antiviral activity [74,75,76,77,78].

It has been seen that *Porphyridium* sp has high activity against murine leukemia virus (MuLV), a single-stranded RNA-enveloped virus from the *Retroviridae* family causing cancer in mice [72,79], and it has interesting anti-HSV activity, reducing infection by 50% and showing no cytotoxic effects in vitro [66,71].

Among the different species, *Porphyridium cruentum* produce some sulphate polysaccharides with high antiviral activity. They have some effects against Varicella zoster (HH3), a double-stranded DNA virus of the *Herpesvirus* family, and VV [71,72]. In a study by Fábregas et al. [80], methanol extracts of this species containing sulphate polysaccharides had high antiviral activity in vitro against two animal viruses, Piscine novirhabdovirus (VHSV) and African swine fever virus (ASFV). VHSV is a single-stranded RNA-enveloped virus from the Rhabdoviridae family causing serious viral hemorrhagic septicemia (*VHS*) and death in salmonid fishes [80]. ASFV is an enveloped double-stranded DNA virus in the *Asfarviridae* family with a high mortality rate and causing a hemorrhagic fever, African swine fever (ASF), in domestic pigs [80].

Two dinoflagellates produce some sulphate polysaccharides with antiviral activity: *Gyrodinium impudium* and *Cochlodinium polykrikoides.*

*Gyrodinium impudium* produces a sulphate polysaccharide rich in galactose and glucose called p-KGO3, and this inhibits IAV until six hours after the viral infection. It has some effects on Encephalomyocarditis virus (EMCV) [81], a non-enveloped single-stranded RNA virus of the *Piconaviridae* family, which is the causative agent of myocarditis and encephalitis, neurological diseases, reproductive disorders and diabetes in many mammalian species [81].

*Cochlodinium polykrikoides* produces a sulphate polysaccharide composed predominantly of mannose, galactose, glucose and uronic acid, with antiviral effects and no cytotoxicity effect against HIV1, IBV, HSV, MuV and parainfluenza viruses (HPIVs) [82], which are single-stranded viruses of the *Paramyxoviridae* family causing severe respiratory disease in children under 5 years of age [83].

One type of diatom collected in deep seawater, *Navicula directa*, produces a sulphate polysaccharide called Naviculan, which is rich in fucose, xylose, galactose, mannose, rhamnose, and has other trace amounts of sugar moieties. It showed its activity in vitro against HSV, IAV and HIV [66].

Lastly, sulphate polysaccharides of green microalga *Chlorella autotrophica* have shown some antiviral effects against animal viruses such as VHSV and ASFV [80,81,82,83,84].

Not all the polysaccharides of microalgae showing anti-HSV activity are sulphated, but some of them are acid polysaccharides with carboxyl group, phosphate group or ester group.

Nostoflan is one of the most famous acidic polysaccharides. It is extracted from the cyanobacterium *Nostoc flagelliforme*. It shows a high activity against HSV: in the presence of this substance, there is a reduction in gD, a major component of the virion envelope glycoprotein thought to be required for the fusion of the virus [85,86]. Nostoflan also has some effect against IAV in vitro, reducing the virus yield at low concentrations, and in vivo in mice models when injected intranasally [87]. These acid polysaccharides did not show antithrombin activity, a typical collateral effect of some sulphate polysaccharides.

*Chlorella pyrenoidosa* produced an acid polysaccharide with some effects in mice against vesicular stomatitis virus (VSV) [17,18,19,20,21,22,23,24,25,26,27,28,29,30,31,32,33,34,35,36,37,38,39,40,41,42,43,44,45,46,47,48,49,50,51,52,53,54,55,56,57,58,59,60,61,62,63,64,65,66,67,68,69,70,71,72,73,74,75,76,77,78,79,80,81,82,83,84,85,86,87,88], a negative stranded enveloped RNA virus of the *Rhabdoviridae* family causing mucosal vesicles and ulcers in the mouths of cattle, horses and pigs [88].
antibiotics-10-00746-t002_Table 2Table 2Microalgae polysaccharides with antiviral effects.Polysaccharides Interact with the Positively Charged Domains of the Virus Envelope Glycoprotein and Create a Non-Reversible ComplexSulphate Polysaccharides:**Name****Organism****Virus****References**Calcium-spirulan (Ca-SP)*Spirulina*HIV1, HIV2, HSV1, HSV2, HCMV, MuV, IAV [20,66]Naviculan*Navicula directa*HSV, IAV, HIV
TK-V3 *Spirulina platensis*HSV1, ECTV, VV[72]/*Arthsospira fusiforme*HSV1, HSV2 [73]/*Porphyridium* sp.HSV, MuLV, [72,74,75,77,78,79] /*Porphyridium cruentum*HH3, VV, ASFV, VHSV [80]/*Cochlodinium polykrikoides*HIV1, HSV, IBV, HPIVs, MuV[81]KGO3*Gyrodinium impudium*IAV, EMCV, [82,83]/*Chlorella autotrophica*VHSV, ASFV[80]Acid Polysaccharides


Nostoflan*Nostoc flagelliforme*HSV, IAV[84,85,86]
*Chlorella pyrenoidosa*VSV[88]

### 2.3. Pigments

Microalgae pigments are extensively used in the biomedical field, and several studies show they could be used similarly to antivirals. Among these, the main substances that showed the most interesting results are pheophorbide a, carotenoids, astaxanthin, and phycobiliproteins (allophycocyanin, phycocyanin) (Table 3).

Pheoporbide a (PPba) is formed after the dephytylation and demetallation of chlorophyll a [89]. This substance, known for its antiproliferative activity and used as an anticancer cure [90], shows some antiviral effects, in particular against enveloped viruses [91]. However, its action is not completely clear. It is commonly believed to bind to virus cell receptors, but some studies demonstrate that it also has some effects in post-entry steps [92,93]. In more detail, cyanobacterium *Lingbya* and green microalga *Dunaliella*, in particular the species *Dunaliella primolecta*, are active against HSV1, and inhibit the cytopathic effect of the virus [94]. The presence of pheophorbide as an antiviral compound was confirmed by NMR and MR analysis.

Carotenoids are used in different biotechnology fields, and their antiviral action could be both direct and indirect. It has been seen that extracts of *Dunaliella salina* directly inhibit the plaque formation of *Suid herpesvirus* (su-HV1), an enveloped DNA virus of the *Herpesviridae* family, causing fever and high mortality in piglets [95].

However, it is also known that carotenoids decrease the harmful effects of some viruses indirectly. Indeed, viruses increase the level of reactive oxygen species (ROS) and reactive nitrogen oxygen (RNS), which inhibit virus replication. However, an excessive production of these substances activates transcription nuclear factor-KB (NF-KB), inducing the jak/stat pathway. Consequently, the cytokine is produced at higher levels, and this could cause a fatal attack by the immune system, called a cytokine storm. This hyperinflammation of the cytokine storm is also the origin of two severe syndromes: acute respiratory distress syndrome (ARDS), causing breathlessness and rapid heart rate, and acute lung injury (ALI), involved in the damage of different tissues. One is the consequence of the other: firstly, the alveolar walls are damaged, and consequently, there is a scarcity of oxygen in other organs [96].

Li et al. [97] showed that the beta-carotene of *Dunaliella salina* inhibits the level of nitric oxide and cytokine; moreover, it downregulates the gene and protein expression of the jak/stat pathway in the presence of pseudorabies virus *(PRV)*, an *Alphaherpes* virus creating neurological problems in mice.

Furthermore, astaxanthin, a xanthophyll produced at high quantities by *Hematococcus pluvialis* and some diatoms [96], showed antioxidant effects, in particular against the collateral effect of *Whispovirus* (white spot syndrome virus, WSSV), a double-stranded enveloped DNA virus of the *Nimaviridae* family that causes white spot disease (WSD), a lethal syndrome in penaeid shrimps, and against infectious hematopoietic necrosis virus (IHNV), a single-stranded DNA virus of the *Rhabdoviridae* family [54,73,98]

Phycobiliproteins are other substances with antiviral activity.

*Spirulina platenis* extracts containing allophycocyanin (APC) were able to delay the RNA synthesis in vitro of *Enterovirus 71* (EV71), a single-stranded RNA virus of the *Piconaviridae family*, causing neurological and cardiovascular disorders [99].

A cold-water extract of *Spirulina platensis* showed anti-IAV activity. Its action is probably linked to the presence of phycocyanin (CPC), which generally downregulates the expression of inflammatory factors caused by viruses [100,101,102].

Some methanolic extracts of *Spirulina* or A*nkistrodesmus convolotus* showed an anti-Epstein Barr virus (EBV) effect, inhibiting some proteins, such as LMP1, EBNA and ZEBRA, linked to the lytic cycle of the virus. This activity is probably linked to unidentified pigments of the phycobiliproteins family [103]. EBV is a member of a *herpesvirus* family that is generally asymptomatic, but, in some cases, it leads to lymphoproliferative disorders, such as Burkitt’s lymphoma (BL) and Hodgkin’s lymphoma (HL) [104].
antibiotics-10-00746-t003_Table 3Table 3Microalgae pigments and derivatives with antiviral effects.NameOrganismVirusActionReferences**Pheophorbide a***Dunaliella**Primolecta*, *Lyngbya*HSV1Bonds to virus cell receptors, effects post-entry steps[89,90,91]**Carotenoids extracts***Dunaliella salina*Su-HV1, PRVInhibition of plaque formation and downregulation of gene and protein expression[94,95,96]**Astaxanthin***Haematococcus pluvialis*WSSV,IHNVAntioxidant action[96,98]**Allophycocyanin***Spirulina platentis*EV71 Delay of viral RNA synthesis in vitro [99]**Phycocyanin***Spirulina platentis*IAVDownregulation of expression of inflammatory factors[102]**Not identified pigment***Ankistrodesmus convolotus*,*Spirulina*EBVInhibition of some proteins involved in the lytic cycle[103,104]

### 2.4. Others Microalgae Compounds with Antiviral Effects

#### 2.4.1. Peptides and Proteins

*Spirulina maxima* produced a peptide called SM that showed antiviral activity against HIV. This peptide inhibits the reverse transcriptase of the virus and p24 antigen production [105].

Other peptides with antiviral activity are cyclic depsipeptides (CDPs), which are particular peptides wherein one or more amino acids are replaced with a hydroxylated carboxylic acid (Table 4).

These peptides are generally synthesized by non-ribosomal peptides in combination with fatty acids in algae, sponges, and other marine organisms [106]. Zainuddin et al. [107,108] showed that ichthyopeptin A, a CDP produced by *Microcystis ichthyoblabe*, had a high IAV activity inhibiting the proteins of the virus cycle.

Different works in the field of aquaculture showed that proteins can also act against viruses.

Studies using HPLC showed that the antiviral effect against nuclear polyhedrosis virus (NPV), a double-stranded DNA-enveloped virus from the *Baculoviridae* family killing the silkworm *Bombyx mori*, is linked to the presence of some proteins produced by *Spirulina platensis*. However, the action is not known [69].

Proteins of *Nannochloropsis oculata* increase α actin activity and immunity system defense, decreasing the mortality linked to Betanodavirus (or nervous disease virus, NNV), an RNA non-enveloped virus from the *Nodaviridae* family causing the nerval necrosis of the humpback grouper [109].

#### 2.4.2. Flavonoids and Polyphenols

Flavonoids are potent antivirals [110] and some of them are produced by microalgae (Table 4). In different experiments, a methanol extract of the cyanobacterium *Geitlerinema* sp. strain containing a substance of the flavonoids group showed high anti-HCV activity, reducing the ATPase activity and consequently RNA helicase and virus replication [110,111,112].

A pigment with a probable polyphenolic nature that shows interesting antiviral activity is marennine [113]. Water-soluble marennine is a blue-grey accessory pigment produced by the tychopelagic diatom *Haslea ostrearia* during blooms. The pigment’s name is derived from the French region Marennes-Olèron, a place rich in this diatom [114,115,116,117].

Marennine is produced in high quantities in a photobioreactor in the presence of shear stress [117], and it is utilized for different industrial purposes, such as in food, colorants and cosmetics [114,118]. Olicard et al. [119] showed that it has effective antiviral activity against HSV and HIV; its action is not clear in detail.

#### 2.4.3. Glycolipids

Two glycolipids, monogalactosyldiacilglyceride and sulfoquinovosyldiacyglycerol, showed an antiviral effect (Table 4).

Monogalactosyldiacilglyceride is produced by *Coccomyxa* sp., a green microalga containing more than 30% lipids by dry weight [120]. This substance is generally the main component of chloroplasts and other organelles. It is a potent viricide of HSV2 in animal cells and in vitro cells. The mechanism of action is not clear: it probably has the ability to change the shape of virus particles, thus harming the viral envelope, and could cause lysis and thus prevent the formation of plaques [121].

Sulfoquinovosyldiacyglycerol is produced by two cyanobacteria, *Phormidium* sp. and *Lyngbya* sp. It is a glycolipid that is rich in sulfur, is associated with photosynthetic membranes, and has high antiviral activity, in particular against HIV and HSV [122,123]. This substance inhibits DNA polymerase thanks to the negatively charged side of the sulphonate group in quinovose, a dioxide carbohydrate, which interacts with the positively charged side of the enzyme (Figure 3).

Consequently, this enzyme cannot interact with the negative charge of phosphodiester bonds [122,123,124].
antibiotics-10-00746-t004_Table 4Table 4Other microalgae compounds with antiviral effects.NameTypologyOrganismVirusActionReferences**Ichthyopeptin A**
Peptide
*Microcystis**ichthyoblabe*
IAV

Inhibition of proteins of virus cycle
[107,108]**SM**
Peptide
*Spirulina maxima*
HIV1

Inhibition of the reverse transcriptase of the virus and p24 antigen production
[105]**Not identified protein**
protein
*Nannochloropsis oculata*
NNV

Increase in α actin activity and immunity system
[109]**Not identified protein**
protein
*Spirulina platensis*
NPV

Decrease in mortality
[105]**Not identified flavonoids**
flavonoid
*Geitlerinema* sp.

HCV

Reduction in ATPase activity
[111,112]**Marennine**
polyphenol
*Haslea ostrearia*
HSV,HIV

Inhibition of virus invasion and replication.
[113,114,115,116,117,119]**Monogalactosyldiacilglyceride**
glycolipid
*Coccomixa* sp.

HSV2

Change of virus shape causing lysis.
[122]**Sulfoquinovosyldiacyglycerol**
glycolipid
*Phormidium* sp.
*Lyngbya* sp.

HIV, HSV

Inhibition of RNA polymerase.
[123]

## 3. Antiviral Bioengineering Perspectives Using Microalgae

Significant assistance in the fight against viruses is offered by the genetic engineering techniques widely used in microalgae, considered as model organisms [125,126,127] (Table 5).

One of these is the recombinant antibodies technique that allows the production of cloning antibodies in a vector, which are then subsequently expressed in a host.

Hempel et al. [128,129] used *Phaeodactylum tricornutum* to treat hepatitis B virus (HBV). In the experiment, monoclonal recombinant human antibody CL.4mAb, which acts against the HBV surface proteins, was expressed and assembled in the endoplasmic reticulum of this microorganism, showing a direct effect against the virus in vitro in human cells.

Genetic engineering is used to create vaccines, which are fundamental weapons against viruses.

The main strategy used to create vaccines, in particular against Sars-CoV 2, is based on a viral vector replicating or not replicating DNA and RNA [130].

An example is the technique of RNA interference (RNAi), wherein a double-stranded RNA interferes with virus mRNAs.

*Chlamydomonas reinhardtii*, a green microalga, is used as a vector to express double-stranded RNA in yellow head virus (YHV), a single-stranded RNA virus of the *Roniviridae* family infecting shrimps and prawns. It is given to animals by oral introduction. The organisms treated with these bioengineered microalgae survive the infection of this virus. We have observed that this virus is related to the family of coronaviruses [131].

Other interesting strategies derived from microalgae could be used to create vaccines.

Reddy et al. [132] tried to create a vaccine against infectious bursal disease virus (IBDV), a non-enveloped RNA virus of the Birnaviridae family causing an immunosuppressive disease in poultry [79,130] thanks to the transformation technique. This method allows the introduction of genetic material into a host organism thanks to the support of competence bacteria. The protein VP2 was expressed in *Chlorella pyrenoidosa* via *Agrobacterium tumefaciens*; serotype-specific antigenic determinants, located on this protein, induce neutralizing antibodies, and confer protection in young chicken against the action of the virus (Figure 4).

Màrquez-Escobar et al. [133] tried to create a vaccine against Zika virus (ZIKV), a single-stranded enveloped RNA virus of the Flaviviridae family causing fetal neurodevelopmental defects through microalgae. An antigenic protein of the envelope of ZIKV was expressed in *Schizochytrium* sp. with the help of *Escherichia coli.* These modified microalgae were administered orally to mice that then showed a high antiviral response.
antibiotics-10-00746-t005_Table 5Table 5Genetic engineering.Genetic EngineeringMicroalgaeUsesVirusReferences*Phaeodactylum tricornutum*Expression of a recombinant antibodyHBV[128,129]*Chlamydomonas reinhardtii*Expression of RNA interferingYHV[131]*Chlorella pyrenoidosa*Expression of antigenic proteinIBDV[132]*Schizochytrium* sp.Expression of antigenic proteinZIKV[133]

## 4. An Overview of the Antiviral Effects of a Supplementary Microalgae Diet and Its Possible Action on SARS-CoV-2

The oral introduction of both microalgae and microalgae compounds is widely used to counteract virus diseases as well as cancers. A diet rich in these substances is not invasive and does not have side effects [134].

One of the best candidates for nutritional properties is Spirulina.

A supplementary diet rich in Spirulina could have some antiviral action thanks to its high content of nutraceutical products.

Some of its derivatives are commercially in use as dietary supplements. For example, the company ChromaDex produces an extract called Immulina [135,136,137], with some action against IAV thanks to its Braun-type lipoproteins. These lipoproteins activate toll-like receptors, and consequently the immunity system [137].

A diet rich in Spirulina is overall known for its beneficial effects against HIV [105,133,138].

Teas et al. [139] noted that there is a difference in the rate of HIV/AIDS incidence among Asia, Chad, and the parts of Africa. This phenomenon was thought to be linked to the consumption of Spirulina, which is very high in Chad and in Asia.

Ngo-Matip et al. [140] showed that a nutrition rich in Spirulina has some effects against the collateral effect of HIV. It increases insulin sensitivity thanks to the antioxidant effects of phycobiliproteins. So, interleukin 6 (IL-6), which inhibits insulin signaling molecules such as insulin receptor substrate, is regularized [141,142,143], and the activity of the lipoprotein lipase, an important enzyme in the lipid metabolism when it is altered in HIV patients, is increased [144,145].

In a study carried out in Cameroon and Burkina Faso [146], based on a randomized multicenter trial, on undernourished children, it was shown that a diet rich in this microalga increased the production of leukocytes, decreasing the probability of developing AIDS.

In a study by Yakoot and Salem [147], there was a decrease in virulence of HCV and of alanine aminotransferase (ALT) in sixty-six patients with chronic hepatitis after three months of treatment. As in the case of HIV, this effect could be linked to the presence of some compounds, such as fatty acids. Generally, fatty acids increase the immune system cell number, and the effects of a diet rich in these substances against different types of viruses were seen in different studies [148].

A diet rich in fatty acids of *Chlorella* increases the immunity system of people aged 50–55 after influenza vaccination, and of *Salmon trutta caspius* after infection with NNV [109].

Therefore, in the case of SARS-CoV-2, a diet rich in fatty acids could have a threefold effect (Figure 5). Firstly, it could increase the amount of T cells targeting spike proteins of SARS-CoV-2 [149]. Secondly, it could help to disintegrate the viral particle by entering the virus’ lipid membrane and destabilizing the bilayer of the envelope; this way, coronavirus’ replication in vitro is suppressed [150,151]. Thirdly, it could help to prevent collateral effects in vaccinated people (Figure 5).

Moreover, Spirulina and Chlorella are rich in other substances with antioxidant effects, such as vitamins and phenols [152,153].

*Hematococcus pluvialis* is rich in antioxidant substances, in particular astaxanthin. Choi et al. [154] showed that a nutrition rich in astaxanthin really reduce ALI and ARDS (Figure 6). So, such a diet could be important in preventing the cytokine storm, a collateral effect of SARS-CoV-2 attack, confirmed by the higher presence of IL-6 in SARS-CoV-2 non-survivors [96,99].

Park et al. [155] showed that a diet supplemented with astaxanthin has an immune booster effect, increasing the number of lymphocytes in peripheral blood cells. This activity could also help to contrast the decrease in lymphocytes and granulocytes in the peripheral blood of people with severe SARS-CoV-2 [96].

A diet containing the green microalga *Chlamydomonas reinhardtii* improves gastrointestinal health thanks to the presence of high-phenol compounds in its biomass [156,157]. So, this kind of diet could also act against another collateral effect of SARS-CoV-2, the gastrointestinal symptoms caused by alterations in the gut microbiome, a symptom reported in 20% of people affected by this disease [158,159].

Furthermore, a diet rich in polysaccharide-inhibiting viruses that bind to the entry site could help to avoid or weaken a viral attack.

It has been shown that the Carrageenan and Chitosan polysaccharides produced by two different macroalgae have some anti SARS-CoV-2 effects [159,160,161]. Carrageenan is produced by the macroalga *Chondrius crispus*, and it is commercially used in a nasal spray solution (Bisolviral^®^) to contrast IAV, but also as a food additive (approved by GRAS) [159,160,161].

Chitosan is produced by *Macrocystis Pyrifera*, and it is also used as a food additive for cholesterol regulation [159,160,161,162,163].

Immurella, a mix of extracted polysaccharides from Chlorella, had interesting effects when supplemented into the diets of broilers [48,137,152,164].

Reichert [165] showed that the mortality linked to Koi herpes virus (KHV), a double-stranded DNA virus of the *Alloherpesviridae* family causing a high death rate in koi carps, decreases in the presence of a diet rich in *Spirulina platensis* exopolysaccharides.

## 5. Conclusions

The SARS-CoV-2 pandemic has demonstrated the necessity of investing in research into new antiviral compounds. Microalgae are considered good candidates to this end. In line with European objectives, they display eco-friendly and eco-sustainable characteristics, produce a high variety of antiviral compounds, and can be used as a supplement in diets without collateral effects. Moreover, these organisms are considered very good candidates for the genetic engineering approach, giving rise to interesting results in recombinant nucleic acid fields that are the basis for new types of vaccines, such as the SARS-CoV-2 vaccines.

## Figures and Tables

**Figure 1 antibiotics-10-00746-f001:**
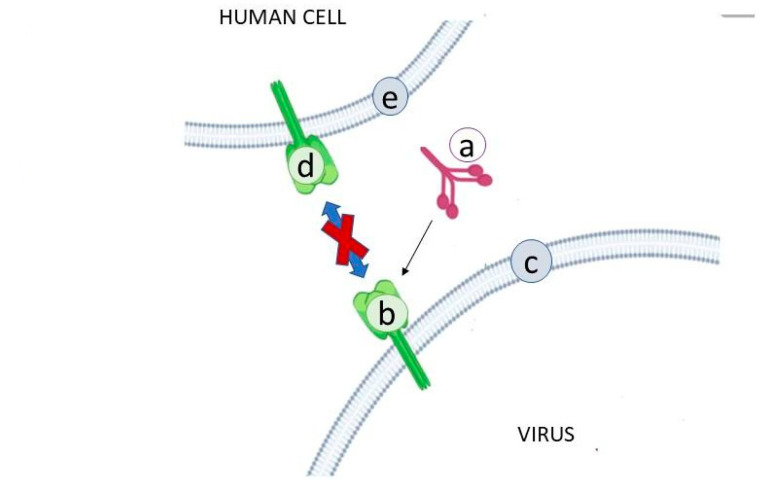
Interaction of the lectin with the virus. The lectin (**a**) interacts with the high-glycan structure (**b**) of viral envelope glycoproteins (**c**), blocking fusion with the glycan structure (**d**) of human cell membrane glycoproteins (**e**).

**Figure 2 antibiotics-10-00746-f002:**
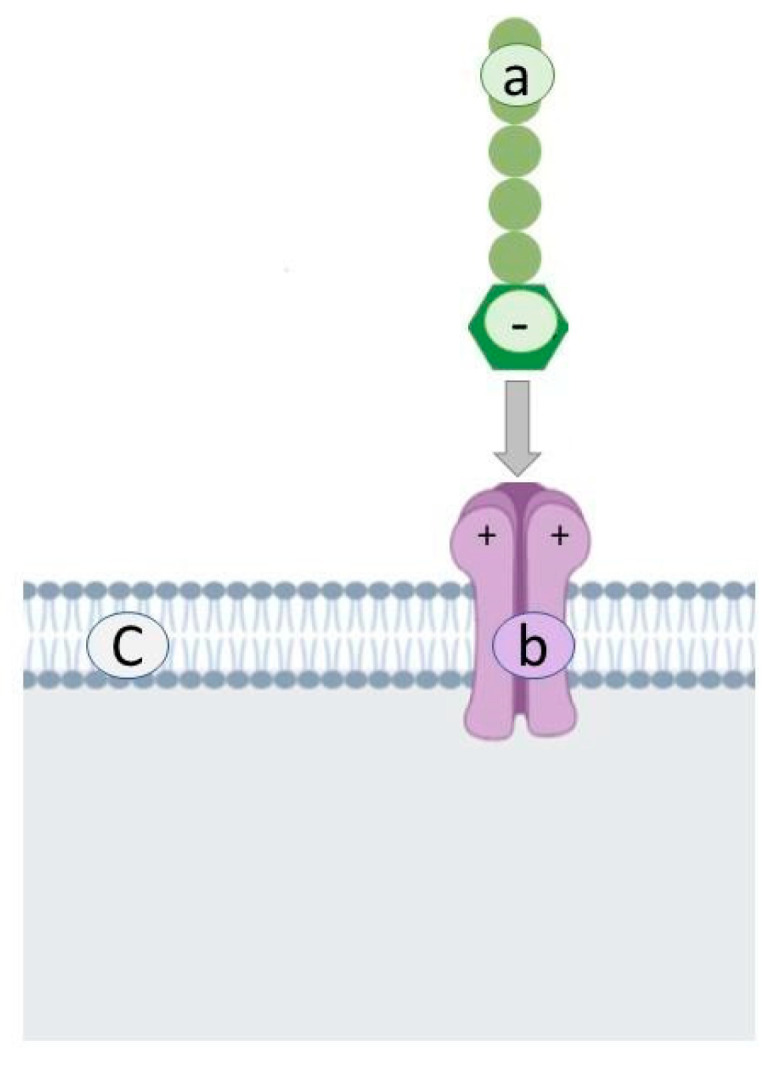
Mechanism of interaction of sulphate polysaccharides. Sulphate polysaccharide (**a**), thanks to the negative charge of the sulfonate group, interacts with the positively charged domains (**b**) of the viral glycoprotein envelope (**c**) and creates a non-reversible complex.

**Figure 3 antibiotics-10-00746-f003:**
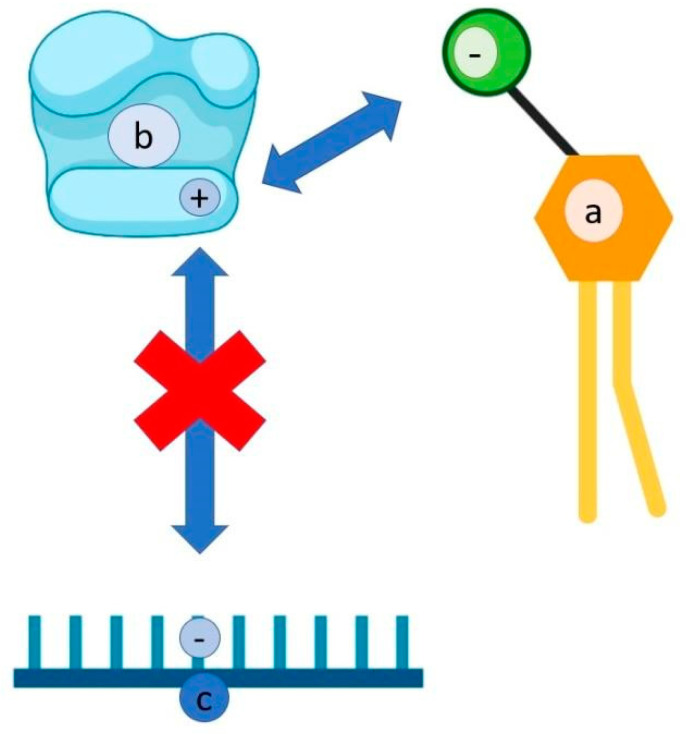
Mechanism of interaction of Sulfoquinovosyldiacyglycerol. Sulfoquinovosyldiacyglycerol, thanks to negative charge (**a**), interacts with the positive charge of DNA polymerase (**b**), thus inhibiting the negative charge of the phosphodiester bonds of DNA (**c**).

**Figure 4 antibiotics-10-00746-f004:**
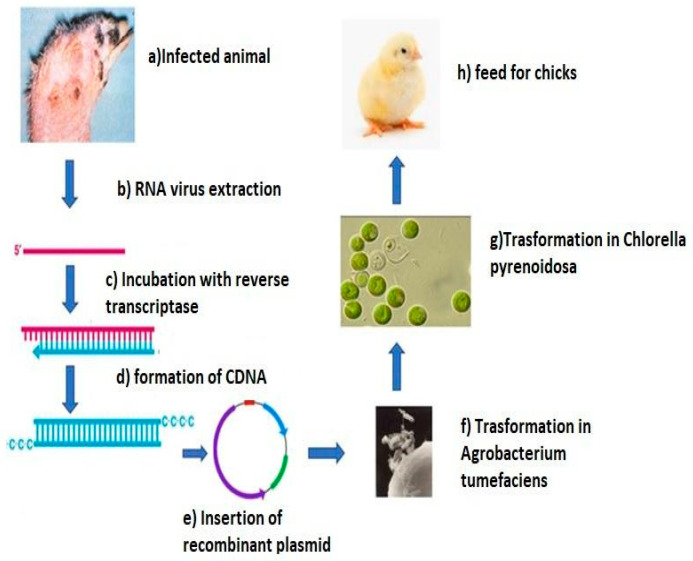
Experiment of Reddy et al. [132].

**Figure 5 antibiotics-10-00746-f005:**
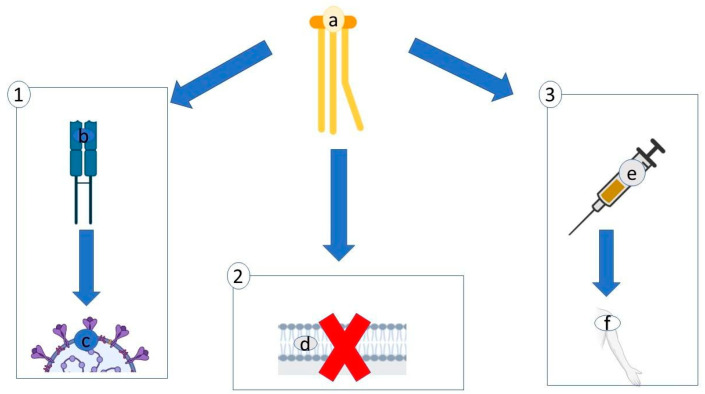
Three potential effects of fatty acids (**a**) against SARS-CoV-2: (1) FA could increase T cells (**b**) that target the spike proteins (**c**); (2) FA penetrates the lipid membrane of the virus, destabilizing the envelope’s architecture (**d**); (3) FA could help to prevent collateral effects in vaccinated people (**e**–**f**).

**Figure 6 antibiotics-10-00746-f006:**
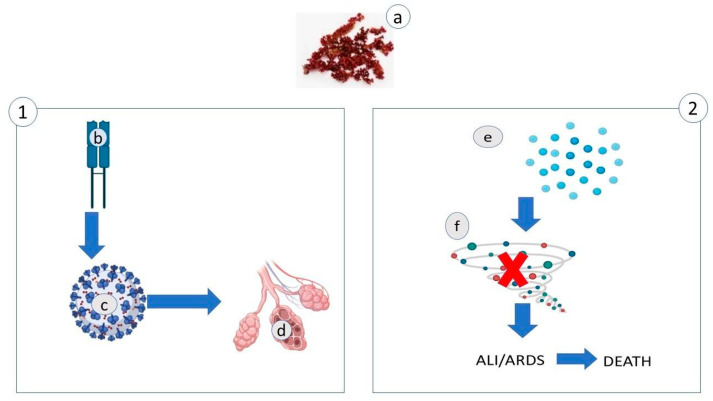
A diet rich in astaxanthin (**a**) has two possible actions against SARS-CoV-2. First action (1): AST increases the number of lymphocytes (**b**) in the presence of coronavirus (**c**) and consequently decreases the oxidative damage of the alveolus (**d**). Second action (2): AST decreases a specific cytokine (such as IL-6, **e**), thus preventing cytokine storm (**f**).

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
