# Peer review of "Evaluation of Microalgae Antiviral Activity and Their Bioactive Compounds"

_antibiotics, 2021, doi:10.3390/antibiotics10060746_

Round 1

Reviewer 1 Report

In this article authors reviewed antiviral activity of microalgae and microalgae derived compounds against common human and animal viruses. Manuscript needs to improve before publication. Specific comments are as below:

1) Manuscript is not stick with the title, it does not deal with the SARS-CoV-2 only where authors also stated this manuscript deal with various  human and animal viruses, thus title need to revise. 

2) Abstract need to revise to highlighted the advancement of the area.

3) Genetic study to develop strain or enhanced target compound in algae has not been research well, however, molecular modeling has been used and many articles have been published in last few years, those can be consider to improve the manuscript. Such as, ‘Algae-Derived Bioactive Molecules for the Potential Treatment of SARS-CoV-2, https://doi.org/10.3390/molecules26082134; Immunoinformatics and molecular modeling approach to design universal multi-epitope vaccine for SARS-CoV-2, https://doi.org/10.1016/j.imu.2021.100578; The COVID 19 novel coronavirus pandemic 2020: Seaweeds to the rescue? Why does substantial, supporting research about the antiviral properties of seaweed polysaccharides seem to go unrecognized by the pharmaceutical community in these desperate times? J. Appl. Phycol. 2020, 32, 1875–1877’, etc.

4) The level of English need to be improve very much. 

5) Line 33-34, ‘Thanks to….living’ this  sentence could be delete.

6) Line 50-53, ‘ Moreover, the treatment…to act’ please consider revising.

7) Line 71-73, ‘In details, we examined….genetic engineering’ sentence is not complete, please consider for revision.

8) Line 412-414, ‘A supplementary diet… fatty acid” please insert reference to support statement.

9) Line 441, Example should be linked with a explanation. Suddenly paragraph can not be started with example. In entire manuscript paragraphs are too short, authors can consider revision to make each paragraph more informative and maintain logical flow of the topic, title, subtitle etc.

Author Response

Note on the revision of the paper: Ms. No. ISSN 2079-6382

Title: Antiviral activity of microalgae compounds and their potential interests against SARS-CoV-2

by Dora Allegra Carbone, Paola Pellone, Claudia Ciniglia, Carmine Lubritto

The Reviewers comments have been carefully considered. The paper has been revised as detailed hereinafter.

Reviewer 1 (highlighted in green)

Comment 1 Manuscript is not stick with the title, it does not deal with the SARS-CoV-2 only where authors also stated this manuscript deal with various human and animal viruses, thus title need to revise.

 Done. The authors modified the title (lines 1-2) and thank the Reviewer for the suggestion

Comment 2 Abstract need to revise to highlighted the advancement of the area.

Done. The authors modified the abstract (lines 14-30) and thank the Reviewer for the suggestion

Comment 3 Genetic study to develop strain or enhanced target compound in algae has not been research well, however, molecular modeling has been used and many articles have been published in last few years, those can be consider to improve the manuscript. Such as, ‘Algae-Derived Bioactive Molecules for the Potential Treatment of SARS-CoV-2, https://doi.org/10.3390/molecules26082134; Immunoinformatics and molecular modeling approach to design universal multi-epitope vaccine for SARS-CoV-2, https://doi.org/10.1016/j.imu.2021.100578; The COVID 19 novel coronavirus pandemic 2020: Seaweeds to the rescue? Why does substantial, supporting research about the antiviral properties of seaweed polysaccharides seem to go unrecognized by the pharmaceutical community in these desperate times? J. Appl. Phycol. 2020, 32, 1875–1877’, etc.

Done. The authors modified the text and add the suggested papers (lines 387-390;472-482,lines 830-831,lines 891-910)

Comment 4 The level of English need to be improve very much.

Done. The authors thank the Reviewer for the suggestion

Comment 5 Line 33-34, ‘Thanks to….living’ this  sentence could be delete.

Done. The authors deleted the sentence and thank the Reviewer for the suggestion

Comment 6 Line 50-53, ‘ Moreover, the treatment…to act’ please consider revising.

Done. The authors changed the text and thank the Reviewer for the suggestion (lines 57-61)

Comment 7 Line 71-73, ‘In details, we examined….genetic engineering’ sentence is not complete, please consider for revision.

Done. The authors changed the text and thank the Reviewer for the suggestion (lines 78-81)

Comment 8 Line 412-414, ‘A supplementary diet… fatty acid” please insert reference to support statement.

Done. The authors inserted the reference and thank the Reviewer for the suggestion (line 445)

Comment 9 Line 441, Example should be linked with a explanation. Suddenly paragraph can not be started with example. In entire manuscript paragraphs are too short, authors can consider revision to make each paragraph more informative and maintain logical flow of the topic, title, subtitle etc

Done. The text was changed and some small paragraphs were eliminated and others were modified

Reviewer 2 Report

This review discussed the antiviral activity of microalgae compounds and their potential applications in the treatment of SARS-CoV-2. Although it is a very interesting, the manuscript is poorly organized, especially there are too many issues with the language. To name a few,

line 2: The usage of "interests" is not accurate, which should be "application"

line 13: "origin" should be "pathogen"

line 15: "prevention" is repetitive.

line 186: "painful" should be "pain"

line 425: "they increase the T cells", confusing. Amount or activity of T cells?

Such mistakes appear all over the manuscript, which is beyond the scope of reversion.

Author Response

Note on the revision of the paper: Ms. No. ISSN 2079-6382

Title: Antiviral activity of microalgae compounds and their potential interests against SARS-CoV-2

by Dora Allegra Carbone, Paola Pellone, Claudia Ciniglia, Carmine Lubritto

Reviewer 2(highlighted in yellow)

Comment 1 line 2: The usage of "interests" is not accurate, which should be "application"

Done. The authors modified the title (line 1) and thank the Reviewer for the suggestion

Comment 2 line 13: "origin" should be "pathogen", line 15: "prevention" is repetitive.

Done. The authors modified the abstract (line 14-30) and thank the Reviewer for the suggestion

Comment 3 line 186: "painful" should be "pain"

Done. The authors modified the text (line 14-30) and thank the Reviewer for the suggestion

Comment 4 line 425: "they increase the T cells", confusing. Amount or activity of T cells?

Done. The authors modified the text (line 449) and thank the Reviewer for the suggestion

Comment 5 Such mistakes appear all over the manuscript, which is beyond the scope of reversion.

            Done. The authors corrected the mistakes in the text

Round 2

Reviewer 1 Report

Manuscript have revised well. However, I found a very interesting review article with the similar topic which can be consider in this paper, this will help to provide more insight to reader. Such as: . Algae-Derived Bioactive Molecules for the Potential Treatment of SARS-CoV-2. Molecules, 26(8):2134 (2021). 

My recommendation is to accept the manuscript upon minor revision. 

Author Response

The Reviewers comments have been carefully considered. The paper has been revised as detailed hereinafter.

Reviewer 1 (highlighted in green)

Manuscript have revised well. However, I found a very interesting review article with the similar topic which can be consider in this paper, this will help to provide more insight to reader. Such as: . Algae-Derived Bioactive Molecules for the Potential Treatment of SARS-CoV-2. Molecules, 26(8):2134 (2021). 

My recommendation is to accept the manuscript upon minor revision. 

Done. The authors add the review (lines 482-490 citation 162-163) and thank the Reviewer for the suggestion.

Reviewer 2 Report

Although this has been improved after reversion, the English language issues are still outstanding. I strongly suggest the authors to ask a third party to polish the manuscript before publication.

Author Response

The Reviewers comments have been carefully considered. The paper has been revised as detailed hereinafter.

Reviewer 2

Although this has been improved after reversion, the English language issues are still outstanding. I strongly suggest the authors to ask a third party to polish the manuscript before publication.

The authors thank Reviewer for the suggestion the english was revised by a native speaker